# Effects of Phlorotannins on Organisms: Focus on the Safety, Toxicity, and Availability of Phlorotannins

**DOI:** 10.3390/foods10020452

**Published:** 2021-02-19

**Authors:** Bertoka Fajar Surya Perwira Negara, Jae Hak Sohn, Jin-Soo Kim, Jae-Suk Choi

**Affiliations:** 1Seafood Research Center, IACF, Silla University, 606, Advanced Seafood Processing Complex, Wonyang-ro, Amnam-dong, Seo-gu, Busan 49277, Korea; ftrnd12@silla.ac.kr (B.F.S.P.N.); jhsohn@silla.ac.kr (J.H.S.); 2Department of Marine Science, University of Bengkulu, Jl. W.R Soepratman, Bengkulu 38371, Indonesia; 3Department of Food Biotechnology, College of Medical and Life Sciences, Silla University, 140, Baegyang-daero 700beon-gil, Sasang-gu, Busan 46958, Korea; 4Department of Seafood and Aquaculture Science, Gyeongsang National University, 38 Cheondaegukchi-gil, Tongyeong-si, Gyeongsangnam-do 53064, Korea

**Keywords:** phlorotannins, toxicity, plant, invertebrates, animal, human

## Abstract

Phlorotannins are polyphenolic compounds produced via polymerization of phloroglucinol, and these compounds have varying molecular weights (up to 650 kDa). Brown seaweeds are rich in phlorotannins compounds possessing various biological activities, including algicidal, antioxidant, anti-inflammatory, antidiabetic, and anticancer activities. Many review papers on the chemical characterization and quantification of phlorotannins and their functionality have been published to date. However, although studies on the safety and toxicity of these phlorotannins have been conducted, there have been no articles reviewing this topic. In this review, the safety and toxicity of phlorotannins in different organisms are discussed. Online databases (Science Direct, PubMed, MEDLINE, and Web of Science) were searched, yielding 106 results. Following removal of duplicates and application of the exclusion criteria, 34 articles were reviewed. Phlorotannins from brown seaweeds showed low toxicity in cell lines, invertebrates, microalgae, seaweeds, plants, animals (fish, mice, rats, and dogs), and humans. However, the safety and toxicity of phlorotannins in aquaculture fish, livestock, and companion animals are limited. Further studies in these organisms are necessary to carry out a systematic analysis of the safety and toxicity of phlorotannins and to further identify the potential of phlorotannins as functional foods, feeds, and pharmaceuticals.

## 1. Introduction

Phlorotannins are polyphenols found in brown seaweeds, consisting of phloroglucinol (Figure 1) (1,3,5-trihydroxybenzene) units that are bonded to each other by different pathways. They are found in the range of 126–650 kDa, and their concentration in dried brown seaweeds varies from 0.5% to 2.5% [1,2]. The concentration of phlorotannins in dried brown seaweeds can differ according to the biological factors, such as age, size, and type of tissue, as well as environmental conditions, such as light intensity, water temperature, season, nutrient levels, and herbivory intensity.

Phlorotannins are classified into four groups based on linkage, such as those with an ether linkage (fuhalols and phlorethols), phenyl linkage (fucols), ether and phenyl linkages (fucophloroethols), and dibenzodioxin linkage (eckols). Numerous phlorotannins have been discovered from brown seaweeds, including eckol [3,4,5], phlorofucofuroeckol A [6,7,8,9], dieckol [2,7,8,9,10,11,12,13,14,15,16,17,18,19,20], 6,6′-bieckol [21,22], 8,8′-bieckol [23], 7-phloroeckol [24], fucodiphloroethol G, phloroglucinol [14], and bifuhalol [25]. Among seaweeds, brown seaweeds including *Ecklonia cava*, have been reported to produce higher concentrations of phlorotannins than other marine phenolic compounds [12]. In brown seaweeds, phlorotannins act as UV protectors and antioxidants, prevent stress and herbivory, and play an important role in the structure of the cell wall [14,15,16].

*Eisenia bicyclis, E. arborea, E. cava, E. kurome, E. stolonifera, Pelvetia siliquosa,* and *Ishige okamurae,* as well as from the genera *Cystophora* and *Fucus*, contain phlorotannins that possess antidiabetic, antioxidant, antitumor, anti-inflammatory, and anticancer properties [26,27,28]. Furthermore, *E. cava* contains eckol, triphlorethol A, fucodiphlorethol G, dieckol, dioxinodehydroeckol, phloroglucinol, and phlorofucofuroeckol A [23], and it exerts antihypertensive [29] and skin-whitening [12] effects. *E. kurome* and *E. bicyclis* were also reported to contain dieckol, phlorofucofuroeckol A, and eckol [30]. An extract of *E. arborea*, which contains phlorofucofuroeckol-B, eckol, and phlorofucofuroeckol A [31], has also shown antiallergic effect. Moreover, the brown seaweed *I. okamurae* contains phloroglucinol and 6,6′-bieckol and diphlorethohydroxycarmalol (DPHC) [32].

Various biological activities have been reported for phlorotannins, including anticancer [23], antioxidant [32], anti-inflammatory [33], antidiabetic, and neuroprotective [18] activities. The biological activities exhibited by these compounds suggest that they are potentially useful as new ingredients in the food [34,35], feed, and pharmaceutical industries. Nonetheless, it is important to study the safety and toxicity of these compounds before the development of new products. In this regard, a review of the safety and toxicity of phlorotannins is urgently needed in view of their functionality and potential for industrial applications. Many articles reviewing the characterization and quantitative analysis of phlorotannin compounds [36,37,38], as well as the functionality of phlorotannins [2,27,39,40,41,42] have been published to date. However, although studies on the safety and toxicity of phlorotannins have been conducted, there are no review papers on this topic. Therefore, this article reviews studies conducted to test the safety and toxicity of phlorotannins in various organisms.

## 2. Materials and Methods

Literature search was conducted systematically based on the Systematic Reviews and Meta-Analyses (PRISMA) guidelines [43]. Online databases, including Science Direct, PubMed, MEDLINE, and Web of Science databases were used. The search strategy focused on the terms “phlorotannins” and “safety OR toxicity OR characteristics OR structure OR cell lines OR microalgae OR seaweed OR plant OR invertebrates OR animals OR human.” Filters included the English language, in vitro, in vivo, and clinical trials. Bibliographies and references from the retrieved records were also considered.

## 3. Results and Discussion

Our literature search resulted 106 articles. After the relevant information was collected and duplicates were removed, 34 articles were reviewed and are summarized in Figure 2. Characteristics and structure of phlorotannins, and the safety and toxicity of phlorotannins in various organisms are described specifically below.

### 3.1. Characteristics and Structure of Phlorotannins

The structure of phlorotannins is formed via dibenzodioxin, ether and phenyl, ether, or phenyl linkages. Fuhalols and phlorethols contain an ether linkage, eckols and carmalols contain a dibenzodioxin linkage, fucols carry an aryl linkage, and fucophlorethols contain an ether and an aryl linkage. An additional hydroxyl group is present in fucophlorethols at the end of the monomer phlorethol unit. Within fucols, the phloroglucinol units can be either linear (as in tetrafucol A) or branched in the meta-position (as in tetrafucol B). Furthermore, heterocyclic fucophlorethols contain dibenzodioxin and furan rings in their structure. Moreover, a 1,4-dibenzodioxin element is present in eckols [1]. The characteristics of phlorotannins isolated from seaweeds are summarized in Table 1.

The classification and characterization of phlorotannins are based on the polymerization of phlorotannins. Increased polymerization increases the structural diversity of phlorotannins. Each class of compounds is known as linear phlorotannins or branched-phlorotannins. Linear phlorotannins (C–C and/or C–O–C oxidative couplings) have only two terminal phloroglucinol residues, whereas branched-phlorotannins have three or more monomer bonds [1]. Parys et al. [48] reported that trifucodiphlorethol A from *Fucus vesiculosus* had six phloroglucinol units containing biphenyl moieties with ortho, ortho′-hydroxyl groups. A symmetrical substitution and two magnetically equivalent protons on rings I, III, and IV were also found, and a symmetrical structure with one fully substituted aromatic moiety (ring II) was confirmed. In trifucotriphlorethol A, seven phloroglucinol units were identified, including 18 hydroxyl groups and a tetraphenyl fragment (rings I–IV) with a non-symmetrical nature in ring II and C-16/C-19 and C-20/C-25 phenoxy-bridges in rings III and V. Furthermore, five units of phloroglucinol were found in fucotriphlorethol A, including 12 acetyl groups and symmetrical substitution of two magnetically equivalent protons on each ring, namely H-2/H-6, H-9/H-11, H15/H-17, H-21/H-23, and H-27/H-29.

Another group of phlorotannins is tetraplorethols A and B, which contain ortho-, meta-, and para-oriented C–O–C oxidative phenolic couplings or combinations thereof [47]. An additional hydroxyl group was found in fuhalols at the end of the monomer [46]. Moreover, a 1,4-dibenzodioxin element has been verified in eckols [45]. To date, 32 phlorotannins have been discovered (see Appendix A) [2,7,14,17,24,25,36,38,39].

### 3.2. Safety and Toxicity of Phlorotannins in Cell Lines

The safety and toxicity of phlorotannins have been evaluated in human [2,3,4,5,11,12,16,17,18,22,24,33,49,50] and animal [13,17] cell lines. The safety and toxicity of phlorotannins in cell lines are summarized in Table 2.

In a study by Quéguineur et al. [49] crude phlorotannins from *A. nodosum* and *Himanthalia elongata* at 0.5–50 µg/mL did not affect the cell viability of HepG2 cells, and were shown to decrease oxidative stress, reactive oxygen species (ROS) generation, and malondialdehyde (MDA) levels, as well as increase glutathione (GSH) concentration, without causing any toxic effects. However, O’Sullivan et al. [33] further observed that the potential cytotoxic effects of crude phlorotannins from *Fucus serratus* not only caused human colon adenocarcinoma (Caco-2) cells to remain viable but also increased GSH activity, decreased deoxyribonucleic acid (DNA) damage, and produced no significant catalase (CAT) activity.

According to Le et al. [17], dieckol from *E. cava* showed low toxicity in KU812 human leukemia cells and RBL-2H3 rat basophilic leukemia cells at 500 μM. After incubation for 4 h, the viability of KU812 and RBL-2H3 cells were 91.76% and 93.66%, respectively. Dieckol suppressed the binding between immunoglobulin E (IgE) and the high affinity IgE receptor (FceRI) receptor, promoting antiallergic mechanism with low toxicity. Lee et al. [18] reported that dieckol extracted from *E. cava* had no effect on the viability of immortalized nontumorigenic human epidermal (HaCaT) cells exposed to particulate matter (PM_10_). Dieckol extracted from *E. cava* attenuated PM_10_ induced cellular lipid peroxidation, and dieckol was effective in attenuating cellular lipid peroxidation both in HaCaT cells and human epidermal keratinocytes. Dieckol also attenuated tumor necrosis factor alph (TNF-α), interleukin-1β (IL-1β), IL-6, and IL-8 expression. Moreover, dieckol extracted from *E. cava* increased the cell viability and decreased prostaglandin E2 (PGE2) release. PGE2 release and gene expression of cyclooxygenase 1 (COX-1), COX-2, and microsomal prostaglandin E synthase-1 (mPGES-1) stimulated by PM_10_ were significantly reduced by dieckol extract [11]. Furthermore, Li et al. [2] reported that dieckol isolated from *E. cava* exhibited no toxic effect in MRC-5 human normal cells, but interestingly, dieckol at the same concentration (400 µM) inhibited the growth of Henrietta Lacks (HeLa), A549, highly tumorigenic (HT1080), and HT29 tumor cells. In addition, dieckol extracted from *E. cava* inhibited 5α-reductase activity in rat vibrissa immortalized dermal papilla cells at 100 µg/mL, as reported by Kang et al. [13], and the proliferation of dermal papilla cells and inhibition of 5α-reductase activity induced by dieckol from *E. cava* led to hair growth, suggesting its potential benefit as a treatment for hair loss. In a study by Ko et al. [16], dieckol from *E. cava* increased the cell viability of HaCaT cells at 100 µg/mL. Dieckol showed a protective effect against ultraviolet B (UVB) radiation-induced cell damage with an 88.42% cell survival rate. Moreover, Heo et al. [12] reported that dieckol from *E. cava* increased the cell viability of human fibroblast cells and decreased UVB radiation-induced damage (tail intensity) and morphological changes in fibroblast cells.

In a report by Zhen et al. [5], eckol from *E. cava* inhibited PM_2.5_^−^ induced generation of intracellular ROS in HaCaT cells, as indicated by 2′,7′-dichlorofluorescein diacetate staining. Eckol also ensured the stability of molecules, maintained a steady mitochondrial state, and protected the cells from apoptosis by inhibiting the mitogen-activated protein kinase (MAPK) signaling pathway. In a study by Lee et al. [4], eckol from *E. cava* did not show any toxic effect in B16F10 melanoma cells. Tyrosinase activity and melanin synthesis in B16F10 melanoma cells were inhibited by eckol. Eckol also reduced the expression of tyrosinase, tyrosinase-related protein 1 (TRP), and TRP2, indicating that eckol is a potent melanogenesis inhibitor. Eckol isolated from *E. stolonifera* inhibited the nuclear factor-kappaB (NF-κB) and activator protein-1 (AP-1) reporter activity in HeLa cells at 10 µg/mL. Inhibition of NF-κB and AP-1 reporter activity may indicate that eckol could protect human skin from collagen degradation by inhibiting matrix metalloproteinase (MMP)-1 expression [3].

According to Zhen et al. [50], diphlorethohydroxycarmalol (DPHC) did not exert any toxic effect in HaCaT cells. DPHC protected the cells against PM_2.5_^-^-induced DNA damage mediated by oxidative stress. DPHC reduced 8-oxoguanine, which was triggered by PM_2.5_^-^. However, Zhang et al. [22] also reported the inhibition of MMP activities; 6,6′-bieckol isolated from *E. cava* did not have major cytotoxic effect, but it reduced cell viability to 96% at a high dose (200 μM). Moreover, 6,6′-bieckol at 100 μM significantly blocked tumor invasion and inhibited the expression of MMP-2 and -9. Interestingly, the levels of NF-κB (p65 and p50) were successfully reduced by 6,6′-bieckol. Furthermore, Yoon et al. [24] reported that 7-phloroeckol isolated from *E. cava* exhibited no toxic effects and inhibited melanin production in B16F10 melanoma cells.

The effects of crude phlorotannins [33,49], dieckol [2,3,11,12,13,16,17,18], eckol [3,4,5], DPHC [50], 6,6′-bieckol [22], and 7-phloroeckol [24] isolated from brown seaweeds, such as *E. cava*, *E. stolonifera*, *I. okamurae*, *F. serratus*, *A. nodosum*, and *H. elongata*, have been examined in cell lines, and they exhibited low toxicity.

Most studies on the safety and toxicity of phlorotannins in cell lines focused only on eckols and dieckol. To the best of our knowledge, the other subclasses of phlorotannins, such as fuhalols, phlorethols, fucols, and fucophloroethols, have not been tested. These phlorotannins have been reported to exert various biological activities, but their safety and toxicity in cell lines have not been reported. Therefore, the action mechanisms of other classes of phlorotannins in cell lines should be further investigated.

### 3.3. Safety and Toxicity of Phlorotannins in Microalgae, Seaweeds, and Plants

The safety and toxicity of phlorotannins extracted from different sources in microalgae, seaweed, and plant have been reported by Nagayama et al. [51], Rengasamy et al. [52], and Cho et al. [53]. The effects of phlorotannins have been tested in microalgae such as *Karenia mikimotoi*, *Cochlodinium polykrikoides*, and *Chattonella antiqua* [52], seaweed such as *Enteromorpha prolifera* [53], as well as plants such as *Vigno mungo* [52]. The safety and toxicity of phlorotannins in microalgae and plants are summarized in Table 3.

According to Nagayama et al. [51], crude phlorotannins from *E. kurome* exhibited toxic effects in several microalgae. The phlorotannins inhibited the swimming activity and increased the mortality (98%) of *K. mikimotoi* within 0.5 h of treatment at 150 mg/L. After treatment at 100 mg/L, 90% of the cells of *K. mikimotoi* changed to non-motile cells within 3 h. In comparison, after administration at 50 mg/L, *K. mikimotoi* cells were still swimming, but the speed was decreased. Furthermore, 100 mg/L crude phlorotannins reduced the cell motility of *C. polykrikoides* by 98% within 0.5 h. Interestingly, the effect of crude phlorotannins was much weaker on *C. antiqua* at 500 mg/L, crude phlorotannins only slightly inhibited the swimming activity of *C. antiqua* within 3 h. In seaweed, crude phlorotannins extracted from *I. sinicola* exhibited strong antifouling activities with no settled spat. At concentration of 30 µg/mL, crude phlorotannins succesfully inhibited the settlement *E. prolifera* spore [53]. In a study by Rengasamy et al. [52], eckol and phloroglucinol isolated from *E. maxima* increased the seedling length and weight of *V. mungo*. Eckol promoted α-amylase activity by increasing the starch clearance zone in the maize scutellum.

Phlorotannins such as eckol, phloroglucinol [52], and crude phlorotannins [51,53] isolated from *E. kurome*, *E. maxima*, and *I. sinicola* exhibited biological activities in microalgae, seaweed, and plant; however, studies on the toxicity of other phlorotannins have not been conducted. The biological activities of these phlorotannins may overcome environmental issues caused by unwanted microalgae, seaweed, and plant. Moreover, to the best of our knowledge, studies on the safety and toxicity of phlorotannins in microalgae and plants are limited. Research on the safety and toxicity of other subclasses of phlorotannins, such as fuhalols, phlorethols, fucols, and fucophloroethols, has not been performed. Therefore, future studies should confirm the safety of phlorotannins in untested microalgae and plants to evaluate the potential of phlorotannins as novel antifouling agents, natural herbicides, and pesticides.

### 3.4. Safety and Toxicity of Phlorotannins in Invertebrates

The safety and toxicity of phlorotannins in invertebrates have been reported by Cho et al. [53], Kim and Choi [54], Lau and Qian [55], and Nagayama et al. [51]. Invertebrates, such as *Portunus trituberculatus* [51], *Artemia salina* [54], *Hydroides elegans* [55], and *Mytilus edulis* [53], have been used to examine the safety and toxicity of phlorotannins. The safety and toxicity of phlorotannins in invertebrates are summarized in Table 4.

According to Nagayama et al. [51], crude phlorotannins extracted from *E. kurome* had no toxic effect on blue crabs (*P. trituberculatus*) after administration at 200 mg/L. After exposure for 0.5 h, *P. trituberculatus* survived, and no side effects were recorded. Furthermore, in a study by Kim and Choi [54], crude phlorotannins extracted from the brown algae *Dictyota dichotoma*, *E. kurome*, *I. okamurae*, *Sargassum sagamianum*, and *Pachydictyon coriaceum* exhibited larvicidal activity against brine shrimp (*A. salina*). The extracts of *D. dichotoma*, *E. kurome*, *I. okamurae*, *S. sagamianum*, and *P. coriaceum* exhibited significant larvicidal activity at 2.5% with a survival rate of 0.0% after treatment for 12, 24, and 48 h; thus, these extracts can be classified as extremely larvicidal.

Lau and Qian [55] showed that crude phlorotannins from *Sargassum tenerrimum* exerted settlement-inhibitory effect on the tube-building polychaete *H. elegans* after treatment at 10^−4^–10^3^ ppm. The LC_50_ of the phlorotannins was 27 times higher than the EC_50_, which were 13.948 ppm and 0.526 ppm. Low EC_50_ values indicated that phlorotannins extracted from *S. tenerrimum* have a potential use as natural settlement inhibitors against polychaetes. In addition, Cho et al. [53] revealed that crude phlorotannins extracted from *Ishige sinicola* and *Scytosiphon lomentaria* had strong antifouling activities on *M. edulis* at 40 µg/10 µL. Crude phlorotannins completely inhibited the repulsive activity of the mussels that were dropped on the foot. The extracts also exhibited strong antifouling activities on larval settlement (6% of the spat settling) at a concentration of 0.8 mg/mL.

Only crude phlorotannins isolated from *D. dichotoma*, *E. kurome*, *I. okamurae*, *S. sagamianum*, *P. coriaceum* [54], *E. kurome* [51], *S. tenerrimum* [55], *I. sinicola*, and *S. lomentaria* [53] have been tested for their safety and toxicity in invertebrates. Considering their excellent biological activities as settlement inhibitors, antifouling agents, and larvicides, crude phlorotannins may be developed as new agents for overcoming the environmental issues caused by the presence of unwanted invertebrates. No other phlorotannins have been evaluated for safety and toxicity in invertebrates. Studies on the effect, concentration, and toxicity of purified phlorotannins in invertebrates have not been conducted. Therefore, further research examining the safety and toxicity of purified phlorotannins in invertebrates should be conducted to evaluate the efficacy of phlorotannins as novel antifouling agents against invertebrates and as pesticides against insects.

### 3.5. Safety and Toxicity of Phlorotannins in Animals

Studies on the safety and toxicity of phlorotannins on animals have been reported in fish, such as seabream (*Pagrus major*), tiger puffer (*Fugu rubripes*) [51], zebrafish (*Danio rerio*) embryos [14], and zebrafish [10], in rodents, such as Institute of Cancer Research (ICR) mice [10,30], HR-1 hairless male mice [5], and Sprague-Dawley (SD) rats [56,57], and in Beagle dog [58]. The safety and toxicity of phlorotannins in animals are summarized in Table 5.

The safety and toxicity of phlorotannins in fish have been reported by [10,14,51]. According to Nagayama et al. [51], crude phlorotannins from *E. kurome* showed minor side effects in *P. major* and *F. rubripes* following treatment at 200 mg/L for 0.5 h. The fish were writhing and gasping for several seconds, after which they calmed down, and some discharged oral mucus. The survival rate of the fish was 100% at the end of the experiment. Kang et al. [14] reported that dieckol isolated from *E. cava* exhibited no toxicity in zebrafish embryos after treatment at 50 μM. Dieckol reduced ROS levels and cell death, inhibited the generation of thiobarbituric acid reactive substances, and elevated survival rate. Heartbeat rate disturbances and apparent adverse effects were not observed during treatment. Moreover, the effect of dieckol in zebrafish has been reported by Choi et al. [10]. After ad libitum feeding with hardboiled egg yolk containing 1 and 4 μM dieckol, body lipid levels in zebrafish were reduced by 40% and 60%, respectively. The levels of the adipogenic factors peroxisome proliferator-activated receptors γ (PPARγ), CCAAT-enhancer-binding proteins (C/EBPα), fatty acid-binding protein 11a (FABP11a), and sterol regulatory element binding factor-1 (SREBF-1) were also decreased by 4 μM dieckol. This result revealed that dieckol inhibited triglyceride accumulation in zebrafish by downregulating adipogenic factors.

Nagayama et al. [30] reported that crude phlorotannins from *E. kurome* showed no toxic effect in ICR mice after treatment for 14 days, showing a survival rate of 100%. The administration of crude phlorotannins at the highest dose did not cause any harmful effects in the mice. Moreover, the weight gain rate in the treated group was not different from that in the control, and no signs or symptoms of disorder were observed. However, Choi et al. [10] reported that dieckol isolated from *E. cava* reduced the final body weight by 35% and body weight gain by 38% in ICR mice after treatment at 60 mg/kg BW/day for 11 weeks. Increased levels of plasma triglycerides, total cholesterol, and low-density lipoprotein cholesterol also decreased after the treatment. High-fat diet (HFD)-induced elevation of liver fat was inhibited, indicating that dieckol prevents the development of fatty liver in response to an HFD. DPHC inhibited lipid peroxidation, protein carbonylation, and epidermal height in HR-1 hairless male mice. Mitogen-activated protein kinase (MAPK) expression in mouse skin was attenuated by DPHC, which indicates that the MAPK signaling pathway may play a key role in PM_2.5_^-^-induced skin damage [50].

In a study by Zaragozá et al. [57], crude phlorotannins from *F. vesiculosus* were shown to lack any relevant toxic effects in SD rats following the administration at 200 mg/kg/day. The crude phlorotannins reduced weight gain in the rats, which showed hematological values within the normal limits for rats and no pathological modification. Moreover, the crude phlorotannins increased α-amylase level to the normal range for rats. Hwang et al. [59] reported that crude phlorotannins from *E. cava* had no toxic effects in SD rats. Treatment with phlorotannins caused no adverse effect on clinical parameters, body weight, ophthalmologic parameters, hematological parameters, coagulation, clinical pathological parameters, and organ weight. All rats survived until the end of both the acute and subacute treatments. Yang et al. [58] reported dieckol from *E. cava* had mild side effects on beagle dogs. During the treatment, one beagle showed soft stool on days 3 and diarrhea on day 13. The survival rate at the end of treatment was 100%.

Phlorotannins compounds, such as dieckol [10,14,58] and DPHC [50], and phlorotannins crude extracts [30,51,56,57] have been tested for their safety and toxicity in animals. However, the toxicity of other phlorotannins compounds has not been studied.

In previous studies, the safety and toxicity of phlorotannins in experimental animals have been extensively investigated using fish, mice, rats, and dogs. Recently, various experimental models have been developed to examine the physiological activities and action mechanisms of bioactive compounds. Thus, the functionality and safety of phlorotannins must be investigated using these new experimental animals. In addition, as phlorotannins have various physiological activities, they are highly likely to be used as feed additives for livestock and companion animals. However, to the best of our knowledge, the in vitro safety and toxicity of phlorotannins in livestock (chicken, duck, swine and cattle) and companion animals (bird and cat) have rarely been reported.

### 3.6. Safety and Toxicity of Phlorotannins in Humans

The safety and toxicity of phlorotannins in humans have been reported by [20,59,60,61]. The safety and toxicity of phlorotannins in humans are summarized in Table 6.

According to Paradis et al. [60], phlorotannins isolated from *F. vesiculosus* and *A. nodosum* had no side effects in 23 participants (12 women and 11 men) following treatment at 250 mg/capsule. They decreased incremental areas under the curve (IAUC) in plasma insulin, post-load plasma insulin concentration, plasma glucose areas under the curve (AUC), and postprandial insulin concentration. Interestingly, phlorotannins from *F. vesiculosus* and *A. nodosum* elevated the level of a surrogate marker for insulin sensitivity.

Baldrick et al. [59] revealed that phlorotannins extracted from *A. nodosum* had no side effect in 80 participants aged 30–65 years following administration at 100 mg/capsule for 8 weeks. Phlorotannins decreased DNA damage and did not significantly improve c-reactive protein (CRP), antioxidant status, or inflammatory cytokines.

In a study by Shin et al. [61], phlorotannins isolated from *E. cava* exhibited no significant adverse effect in 107 participants (138 men and 69 women) following administration at 72 mg and 144 mg/capsule. They decreased the total cholesterol/high-density lipoprotein cholesterol level, body fat ratio, atherogenic index, total cholesterol/low-density lipoprotein cholesterol level, body mass index (BMI), waist circumference, and waist/hip ratio. Furthermore, Um et al. [20] reported that treatment with phlorotannins from *E. cava* in 24 participants showed no serious adverse effects, such as mild fatigue, dizziness, nausea, and abdominal distension. Phlorotannins successfully raised sleep duration scores and inhibited the onset of wakefulness after sleep.

The application of phlorotannins as food supplements and functional food ingredients has been reported by Turck et al. [34] and Catarino et al. [35]. As a food supplement, daily intake of phlorotannins depends on the age of the consumer. For adolescents (12–14 years of age), the maximum daily intake was 163 mg/day. For those above 14 years of age and adults, the daily intake was 230 mg/day and 263 mg/day, respectively. The European Food Safety Authority (EFSA) Panel on Dietetic Products, Nutrition and Allergies (NDA), pursuant to Regulation (EC) No. 258/97, announced that novel food supplements from phlorotannins (marketed as SeaPolynol^TM^) are safe for human consumption [34].

As the number of human diseases reported has increased annually, it is important to find new natural drug agents with low toxicity. Phlorotannins, dieckol, and crude phlorotannins isolated from seaweeds have been shown to decrease plasma glucose and DNA damage as well as increase the sleep duration. Because of their activity and low side effects, these compounds may be developed as new drugs for humans. However, other subclasses of phlorotannins, such as fuhalols, phlorethols, fucols, and fucophloroethols, have not been tested. Therefore, the action mechanisms of other classes of phlorotannins should be further investigated to evaluate the potential of phlorotannins as novel drug agents for humans.

## 4. Conclusions

Recent studies provided evidence that phlorotannins from brown seaweeds showed low toxicity in cell lines (15 articles); microalgae; seaweed spores; plants (3 articles); invertebrates (4 articles); animals [fish, mice, rats, and dogs (8 articles)], and humans (4 articles) at a moderate dosage. Mild side effects were recorded in humans, fish, and dogs. However, in other organisms, there was no toxicity from phlorotannins, which have various biological activities. These findings can be the basis for developing these compounds as novel functional foods, feeds, and pharmaceuticals. To date, to the best of our knowledge, no studies have been performed on the safety and toxicity of phlorotannins in aquaculture fish; livestock (chicken, duck, swine, and cattle); and companion animals (birds and cats). The safety, toxicity, and availability of phlorotannins in these organisms should be verified with further studies.

## Figures and Tables

**Figure 1 foods-10-00452-f001:**
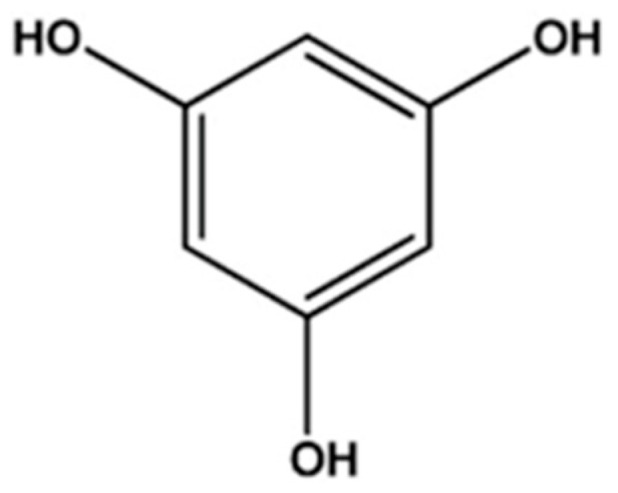
The basic structure of phlorotannins are composed of phloroglucinol units.

**Figure 2 foods-10-00452-f002:**
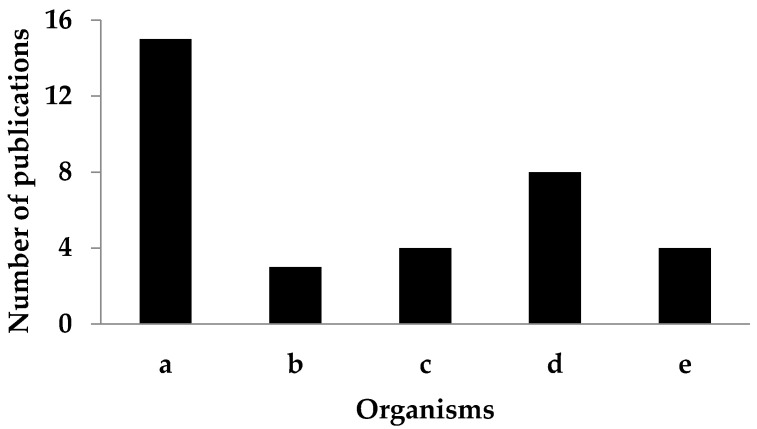
The number of publication related to safety and toxicity of phlorotannins in various organisms, including cell lines (**a**), microalgae, seaweed, and plants (**b**), invertebrates (**c**), animals (**d**), and humans (**e**).

**Table 1 foods-10-00452-t001:** Studies of characteristics and structure of phlorotannins extracted from brown seaweeds.

Phlorotannins	Sources	Administration Method	Characteristics	Ref.
Bifuhalol	*Sargassum* spp. and *Spinuligerum* spp.	^1^H nuclear magnetic resonance (NMR), ^13^C NMR	[3,5-diacetyloxy-4-(3,4,5triacetyloxyphenoxy)phenyl] acetate	[25]
Dieckol	*Ecklonia cava*	^1^H NMR, ^13^C NMR	Two phloroglucin units are cyclized to form diphenyl dioxygen in the dehydrogenated oligomers of three pyrogallol units	[14]
Difucol	*Bifurcaria bifurcata*	^1^H NMR, ^13^C NMR	2-(2,4,6-trihydroxyphenyl) benzene-1,3,5-triol	[44]
Dioxinodehydroeckol	*Ecklonia cava*	^1^H NMR, ^13^C NMR	[1,4] benzodioxino [2,3-a] oxanthrene-1,3,6,9,11-pentayl pentaacetate	[24]
Eckols	*Ecklonia maxima*	MS 50, ^1^H NMR, ^13^C NMR	Contain a 1,4-dibenzodioxin element in their structure	[45]
Eckstolonol	*Ecklonia cava*	^1^H NMR, ^13^C NMR	[1,4] benzodioxino [2,3-a] oxanthrene substituted by hydroxy groups at positions 1, 3, 6, 9, and 11	[14]
Fucodiphloroethol G	*Ecklonia cava*	^1^H NMR, ^13^C NMR	Biphenyl-2,2′,4,4′,6-pentol substituted by a 2,4-dihydroxy-6-(2,4,6-trihydroxyphenoxy)phenoxy substituent at position 6′	[2]
Fucotriphlorethol A	*Ecklonia cava*	^1^H NMR, ^13^C NMR	Has five units of phloroglucinol, including 12 acetyl groups. Having symmetrical substitution of two magnetically equivalent protons on each ring i.e., H-2/H-6, H-9/H-11, H15/H-17, H-21/H-23, and H-27/H-29	[24]
Fuhalols	*Sargassum spinuligerum*	^1^H NMR, ^13^C NMR, heteronuclear multiple bond correlation (HMBC), and mass spectral data	The end monomer unit of phlorethols has an additional hydroxyl group	[46]
Phlorofucofuroeckol A	*Ecklonia cava*	^1^H NMR, ^13^C NMR	1,11-di-(3,5-dihydroxyphenoxy) benzofuro(3,2-a) dibenzo(b,e) (1,4)-dioxin-2,4,8,10,14-pentaol	[2]
Tetrafucol A	*Fucus* spp. and *Dictyotales* spp.	^1^H NMR, ^13^C NMR	2-(2,4,6-trihydroxyphenyl)-4-[2,4,6-trihydroxy-3-(2,4,6-trihydroxyphenyl) phenyl] benzene-1,3,5-triol	[25]
Tetraplorethols A and B	*Laminaria ochroleuca*	Thin-layer chromatography (TLC), ^1^H NMR, and MS data	Ortho-, meta-, and para-oriented C–O–C oxidative phenolic couplings	[47]
Trifucodiphlorethol A	*Ecklonia cava*	^1^H NMR, ^13^C NMR	Has six units of phloroglucinol, containing biphenyl moieties with ortho, ortho -hydroxyl groups. Having a symmetrical substitution and two magnetically equivalent protons on ring I, III, and IV, and a symmetrical structure with one fully substituted aromatic moiety (ring II)	[24]
Trifucotriphlorethol A	*Ecklonia cava*	^1^H NMR, ^13^C NMR	Has seven units of phloroglucinol, including 18 hydroxyl groups. Having a tetraphenyl fragment (rings I–IV) with non-symmetrical nature in ring II and C-16/C-19 and C-20/C-25 phenoxy-bridges in ring III and V	[24]
6,6′-bieckol	*Ecklonia cava*	^1^H NMR, ^13^C NMR	1,1′-bioxanthrene substituted by 3,5-dihydroxyphenoxy groups at position 6 and 6′ and hydroxy groups at positions 2, 2′, 4, 4′, 7, 7′, 9, and 9′, respectively	[17]
7-phloroeckol	*Ecklonia cava*	^1^H NMR, ^13^C NMR	The hydroxy group at position 7 is replaced by a 2,4,6-trihydroxyphenoxy group	[24]

**Table 2 foods-10-00452-t002:** Studies of safety and toxicity of phlorotannins extracted from brown seaweeds, as measured in human and animal cell lines.

Cell Lines	Extract or Chemical	Sources	Method	Concentration	Toxicity	Ref.
HaCaT cells exposed to PM_2.5_^-^	Eckol	*Ecklonia cava*	Cells were cultured with eckol and/or PM_2.5_^-^ for 24 h	30 µM	Not toxic	[5]
HaCaT cells exposed to PM_2.5_^-^	Diphlorethohydroxycarmalol	*Ishige okamurae*	MTT (3-(4,5-Dimethylthiazol 2-yl)-2,5-diphenyltetrazolium bromide) assay and incubated for 16 h	2.5, 5, 10, 20, and 40 µM	Not toxic	[50]
HaCaT cells exposed to PM_10_	Dieckol	*Ecklonia cava*	MTT assay and incubated for 2 h at room temperature	20–100 µg/mL	Decreased cell viability and increased PGE_2_ release	[11]
HaCaT cells exposed to PM_10_	Dieckol	*Ecklonia cava*	MTT assay and incubated for 2 h at room temperature	25, 50, 75, and 100 µg/mL	Had no effects on the viability of HaCaT cells	[18]
HaCaT cells	Dieckol	*Ecklonia cava*	MTT assay and incubated for 4 h	100 µg/mL	Increased cell viability	[16]
HeLa cells	Eckol and dieckol	*Ecklonia stolonifera*	MTT assay and incubated for 4 h	10 µg/mL	Inhibited NF-kB and AP-1 reporter activity	[3]
HeLa, A549, HT1080, HT-29, and MRC-5 (human normal cell line)	Dieckol	*Ecklonia cava*	MTT assay and incubated for 4 h	400 µM	Not toxic on MRC-5 and had activity on tumour suppressive	[2]
Human colon adenocarcinoma (Caco-2) cells	Crude extract	*Fucus serratus*	MTT assay and incubated for 24 h	100 µg/mL	Not toxic	[33]
Human fibroblast cell	Dieckol	*Ecklonia cava*	MTT assay and incubated for 4 h	5, 50, and 100 μM	Increased cell viability	[12]
Human fibrosarcoma cell line (HT1080)	6,6′-bieckol	*Ecklonia cava*	MTT assay and incubated for 4 h	0–200 μM	Not significantly toxic and blocked tumor invasion	[22]
Human hepatoma HepG2 cells	Crude extract	*Ascophyllum nodosum* and *Himanthalia elongata*	Lactate dehydrogenase (LDH) leakage and crystal violet assay	0.5–50 µg/mL	Not toxic	[49]
B16F10 melanoma cells	Eckol	*Ecklonia cava*	MTT assay and incubated for 4 h	25, 50, and 100 μM	Not toxic and inhibited tyrosinase activity and melanin synthesis	[4]
B16F10 melanoma cell	7-phloroecko	*Ecklonia cava*	MTT assay and incubated for 4 h	6.25–100 μM	Not toxic and inhibited melanin production	[24]
Human leukemia cell line (KU812) and rat basophilic leukemia cell line (RBL-2H3)	Dieckol	*Ecklonia cava*	MTT assay and incubated for 2 h at room temperature	500 μM	Low toxicity	[17]
Rat vibrissa immortalized dermal papilla cell line	Dieckol	*Ecklonia cava*	MTT assay and incubated for 4 h	100 µg/mL	Inhibited 5α-reductase activity	[13]

**Table 3 foods-10-00452-t003:** Studies of safety and toxicity of phlorotannins extracted from brown seaweeds, as measured in microalgae, seaweed, and plant.

Microalgae/Seaweed/Plant	Extract or Chemical	Sources	Method	Concentration	Toxicity	Ref.
*Chattonella antiqua*	Crude extract	*Ecklonia* *kurome*	Phlorotannins were dissolved in 70% methanol and were added to 20 mL of microalgal suspensions	20–500 mg/L	Inhibited swimming, and lost their motility	[51]
*Cochlodinium polykrikoides*	Crude extract	*Ecklonia* *kurome*	Phlorotannins were dissolved in 70% methanol and were added to 20 mL of microalgal suspensions	20–500 mg/L	Inhibited swimming, and lost their motility	[51]
*Enteromorpha prolifera*	Crude extract	*Ishige sinicola*	A 1.0 mL aliquot of spore was added to the extract	30 µg/mL	Inhibited settlement of the spore	[53]
*Vigno mungo*	Eckol and phloroglucinol	*Ecklonia maxima*	The seeds were planted in trays and were added eckol and phloroglucinol	10^−3^–10^−7^ M	Increased seedling length and weight	[52]
*Kerenia mikimotoi*	Crude extract	*Ecklonia* *kurome*	Phlorotannins were dissolved in 70% methanol and were added to 20 mL of microalgal suspensions	20–500 mg/L	Inhibited swimming, and lost their motility	[51]

**Table 4 foods-10-00452-t004:** Studies of safety and toxicity of phlorotannins extracted from brown seaweeds, as measured in invertebrates.

Invertebrates	Extract or Chemical	Sources	Method	Concentration	Toxicity	Ref.
*Artemia salina*	Crude extract	*Dictyota dichotoma, Ecklonia kurome, Ishige okamurae, Sargassum sagamianum*, and *Pachydictyon coriaceum*	50 μL of brine shrimp larvae solution (containing ca. 20 larvae) was added with extracts and determined after 2, 4 6, 12, 24, and 24 h of exposure	0.25%, 0.5%, and 2.5%	2.5% extract had significant larvicidal activity	[54]
*Hydroides elegans*	Crude extract	*Sargassum tenerrimum*	Twenty larvae were introduced into each Petri dish containing 5 mL of test solution and were incubated at 28 °C. Survivorship was determined after 48 h of incubation.	10^−4^–10^3^ ppm	EC_50_ at 0.526 ppm and LC_50_ at 13.948 ppm	[55]
*Mytilus edulis*	Crude extract	*Ishige sinicola* and *Scytosiphon lomentaria*	10 µL seaweed extract was dripped on the foot.	40 µg/10 µL	Inhibited the repulsive activity of the foot and strong antifouling activities	[53]
*Portunus trituberculatus*	Crude extract	*Ecklonia* *kurome*	0.5-h exposure	200 mg/L	No died	[51]

**Table 5 foods-10-00452-t005:** Studies of safety and toxicity of phlorotannins extracted from brown seaweeds, as measured in animals.

Animals	Extract or Chemical	Sources	Method	Concentration	Toxicity	Ref.
Seabream	Crude extract	*Ecklonia kurome*	0.5-h exposure	200 mg/L	Light side effects, and no death	[51]
Tiger puffer	Crude extract	*Ecklonia kurome*	0.5-h exposure	200 mg/L	Light side effects, and no death	[51]
Zebrafish embryo	Dieckol	*Ecklonia cava*	Embryos were treated with 25 mM AAPH (2,20-azobis-2-methyl-propanimidamide dihydrochloride) or co-treated with AAPH and phlorotannins for up to 120 h post-fertilization (120 hpf).	50 μM	No conspicuous adverse effects and did not generate any heartbeat rate disturbances	[14]
Zebrafish	Dieckol	*Ecklonia cava*	Feed ad libitum with hardboiled egg yolk as a high-fat diet (HFD) once per day in the presence or absence of the indicated compounds (17–20 dpf) or vehi- cle (dimethyl sulfoxide, DMSO) for 12–15 days (17–20 dpf) for 12–15 days.	1 and 4 μM	Reduced the levels of body lipids	[10]
ICR mice	Crude extract	*Ecklonia kurome*	Oral, free access to food and tap water for 14 days	0, 625, 1250, 2500, 5000 mg/L	No death	[30]
ICR mice	Dieckol	*Ecklonia cava*	Oral, treatment during 11 weeks	60 mg/kg BW/day	Reductions of final body weight and body weight gain	[10]
HR-1 hairless male mice	Diphlorethohydroxycarmalol	*Ishige okamurae*	The mouse dorsal skin was placed in continuous contact with the pads for 7 days	200 µM and 2 mM	Inhibited lipid peroxidation, protein carbonylation, and epidermal height	[50]
Sprague dawley rats	Crude extract	*Fucus vesiculosus*	Oral, treatment during 4 weeks	200 mg/kg/day	Lack any relevant toxic effects	[57]
Sprague dawley rats	Crude extract	*Ecklonia cava*	oral, 1 time/day for 4 weeks	0, 222, 667, 2000 mg/kg BW	No death	[56]
Beagle dogs	Dieckol	*Ecklonia cava*	Oral, treatment during 15 days	750 mg/kg BW	Soft stool and diarrhea	[58]

**Table 6 foods-10-00452-t006:** Studies of safety and toxicity of phlorotannins extracted from brown seaweeds, as measured in humans.

Participants	Extract or Chemical	Sources	Method	Concentration	Toxicity	Ref.
Twenty-three participants (11 men, and 12 women) aged 19–59 years	Crude extract	*Ascophyllum nodosum* and *Fucus vesiculosus*	Oral, 1 capsule/day, treatment during 1 week	250 mg/capsule	No side effect	[60]
Twenty-four participants	Dieckol	*Ecklonia cava*	Oral, 2 capsules/day, treatment during 1 week	500 mg/capsule	No serious adverse effects	[20]
Eighty participants aged 30–65 years	Crude extract	*Ascophyllum nodosum*	Oral, 1 capsule/day, treatment during 8 weeks	100 mg/capsule	No side effect	[59]
107 participants (138 men, and 69 women) aged 19–55 years	Crude extract	*Ecklonia cava*	Oral, 1 capsule/day, treatment during 12 weeks	72 mg/capsule and 144 mg/capsule	No side effect	[61]

## Data Availability

Data supporting reported results are available upon request.

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
