# Peer review of "Effects of Phlorotannins on Organisms: Focus on the Safety, Toxicity, and Availability of Phlorotannins"

_foods, 2021, doi:10.3390/foods10020452_

Round 1

Reviewer 1 Report

The paper „Effects of Phlorotannins on Organisms: Focus on the Safety, Toxicity, and Availability of Phlorotannins“ describes a new and really interesting subject. Phlorotannins have shown some positive effects and the knowledge of these compounds should be investigated further. The paper needs some minor modifications

Names of seaweeds and microorganisms

- L119, 123, 126, 127, 129 and so on - In the text, the names of seaweeds are not written in italic. Check and correct (in Tables the names are italic).

-in the whole text of the manuscript, the names of microorganisms are not italic. Check through the whole text and correct.

Abbreviations

The text has some abbreviations which are not explained when first mentioned. Some of them are well known, but some are not. Some people that are not so familiar with those terms can read the paper. So my strong suggestion is to explain abbreviations when first mentioned (when this is appropriate).

-L 128 - some of abbreviations need explanation when written the first time like dUTP/dTTP

-L 133 - SPR sensorgram – explain the abbreviation

-L207-211 – explain the abbreviation ROS, MDA, GSH, CAT

-L218 – PM10 - explain

-L251 – DPHC – explain abbreviation

L129-130 – „Phlorofucofuroeckol A isolated from E. cava inhibited H1N1 with a 50% effective concentration (EC50) value of 13.48 ± 1.93 μM [6]“ In the Table 2, next to this reference, the concentration is 20 μM. Can you check and correct if necessary?

-169-171 – „In a study by Kim et al., [50], crude phlorotannins showed an inhibition zone of 9.5 ± 0.5 to 15.0 ± 0.1 mm against Listeria monocytogenes with a minimum inhibitory concentration (MIC) of 16 to 32 μg/mL“. Check the reference. In the text, it is a reference 50, and in the Table 3, the reference is 51.

-L171-174 – it describes the minimun inhibitory concentration 2-7.8 mg/ml. The Table 3 mentions different concentrations. Can you check and correct if necessary?

L349-350 – „According to Nagayama et al., [54], crude phlorotannins from E. kurome had no toxic effect in P. major and F. rubripes after treatment at 200 mg/L. All organisms were treated with phlorotannins for 0.5 h. The results showed that the fish writhed with gasping for several seconds or more, and then they eventually calmed down, and some fish discharged oral mucus.“ Check if this is correct because after claiming that there is no toxic effect, you are claiming that fish calmed down and discharged some mucus. Can you explain more. In the Table 7, the result is No died.

-L 375 – check if reference 5 is maybe 53

-L 386 - check if reference 5 is maybe 53

Table 2 should be organized according to human and then animal viruses. And also it would help a reader if the Table follows the description in the text. This can be applied to all tables. If appropriate, it would be better to organize the content of Tables according to the order in the text of the paper.

Author Response

Response to Reviewer 1 Comments

The paper “Effects of Phlorotannins on Organisms: Focus on the Safety, Toxicity, and Availability of Phlorotannins“ describes a new and really interesting subject. Phlorotannins have shown some positive effects and the knowledge of these compounds should be investigated further. The paper needs some minor modifications.

Point 1: L119, 123, 126, 127, 129 and so on - In the text, the names of seaweeds are not written in italic. Check and correct (in Tables the names are italic). In the whole text of the manuscript, the names of microorganisms are not italic. Check through the whole text and correct.

Response 1: Thank you for your comment. We have corrected and checked all scientific names of organisms in the revised manuscript.

Point 2: The text has some abbreviations which are not explained when first mentioned. Some of them are well known, but some are not. Some people that are not so familiar with those terms can read the paper. So my strong suggestion is to explain abbreviations when first mentioned (when this is appropriate).

- L 128 - some of abbreviations need explanation when written the first time like dUTP/dTTP

- L 133 - SPR sensorgram – explain the abbreviation

- L207-211 – explain the abbreviation ROS, MDA, GSH, CAT

- L218 – PM10 – explain

- L251 – DPHC – explain abbreviation

Response 2: In response to your comment, we have spelled out the following abbreviations in the revised manuscript.        

Point 3: L129-130 – “Phlorofucofuroeckol A isolated from E. cava inhibited H1N1 with a 50% effective concentration (EC50) value of 13.48 ± 1.93 μM [6]“ In the Table 2, next to this reference, the concentration is 20 μM. Can you check and correct if necessary?

Response 3: In response to your comment, we have deleted the section on antiviral and antimicrobial activities considering the comment from reviewer 2.

Point 4: 169-171 –“In a study by Kim et al., [50], crude phlorotannins showed an inhibition zone of 9.5 ± 0.5 to 15.0 ± 0.1 mm against Listeria monocytogenes with a minimum inhibitory concentration (MIC) of 16 to 32 μg/mL“. Check the reference. In the text, it is a reference 50, and in the Table 3, the reference is 51.

Response 4: In response to your comment, we have deleted the section on antiviral and antimicrobial activities considering the comment from reviewer 2.

Point 5: L349-350 – “According to Nagayama et al., [54], crude phlorotannins from E. kurome had no toxic effect in P. major and F. rubripes after treatment at 200 mg/L. All organisms were treated with phlorotannins for 0.5 h. The results showed that the fish writhed with gasping for several seconds or more, and then they eventually calmed down, and some fish discharged oral mucus.“ Check if this is correct because after claiming that there is no toxic effect, you are claiming that fish calmed down and discharged some mucus. Can you explain more? In the Table 7, the result is No died.

Response 5: In response to your comment, we have revised to “According to Nagayama et al. [51], crude phlorotannins from E. kurome showed minor side effects in P. major and F. rubripes following treatment at 200 mg/L for 0.5 h. The fish were writhing and gasping for several seconds, after which they calmed down, and some discharged oral mucus. The survival rate of the fish was 100% at the end of the experiment”.

Point 6: L 375 – check if reference 5 is maybe 53.

Response 6: This is an error. We have corrected to “50” in the revised reference section.

Point 7: L 386 - check if reference 5 is maybe 53.

Response 7: This is an error. We have corrected to “50” in the revised reference section.

Point 8: Table 2 should be organized according to human and then animal viruses. And also it would help a reader if the Table follows the description in the text. This can be applied to all tables. If appropriate, it would be better to organize the content of Tables according to the order in the text of the paper.

Response 8: In response to your comment, we have deleted the section on antiviral and antimicrobial activities considering the comment from reviewer 2.”

Thank you for your valuable comments on our manuscript. We have modified our manuscript according to your suggestions and we ask that the revised manuscript be re-examined and considered for publication.

Reviewer 2 Report

Manuscript: foods-1098629

Title: Effects of Phlorotannins on Organisms: Focus on the Safety, Toxicity, and Availability of Phlorotannins

I like the effort of the authors to compile the available information about safety and toxicity of these phenolic compounds, which have been reported to exert many biological activities. Lately, numerous scientific works have stated that compounds extracted from macroalgae may be use in different industrial applications, thus, evaluate their safety is very necessary.

In the following paragraphs, I will provide clear information in order to improve the manuscript. In general, the English level is good and I cannot see any discrepancies. I have detected that italics is missing in many species names throughout the manuscript, so I suggest the authors to carefully check these mistakes.

  1. Introduction
  • The first section is not fluid, the authors used too much “phlorotannins”.
  • Line 47: “higher phenolic compounds”??
  • Line 48: In my opinion, it would be interesting to explain a bit more the function of phlorotannins in brown seaweeds.
  • Lines 51-60: Is there a reason why the authors focus on the species Eisenia bicyclis, E. arborea, E. cava, E. kurome, E. stolonifera, Pelvetia siliquosa, and Ishige okamura? There are phlorotannins extracted from other brown macroalgae which have been reported to exert biological properties.
  • Line 51: and

In my opinion, this section is not enough focused regarding the objective presented by the authors. It is too general. Authors mentioned have mentioned food, pharmaceutical and cosmetic products in the abstract, but in the introduction they do not mention the need to evaluate the toxicity of phlorotannins for such applications. In addition, I would also suggest to remark the possibilities offered by phlorotannins in the field of food, so that the manuscript fits better with the scope of the journal.

  1. Results and discussion

I have several comments regarding this section. Since the review aims to compile the information about the safety and toxicity.

  • Is there any relationship between toxicity/safety and the structure of the phlorotannins?
  • Why authors have reviewed antiviral and antimicrobial properties? Is interesting information, but I’m not sure if it fits with the objective of the manuscript.
  • Line 206: Himanthalia elongata
  • In Paragraph 3.5. Safety and Toxicity of Phlorotannins in Microalgae, Seaweeds and Plants, no species of seaweeds (=macroalgae) is mentioned in the text or the table, so I would recommend to change the title.
  • I would modify Figure 2, according with previous suggestions, and introduce it at the beginning of the section.
  1. Conclusions

From my point of view, authors should elaborate a bit more their conclusions. They have compiled interesting information, but they do not draw elaborate conclusions.

FINAL REMARKS

In my opinion, I am suggesting MAJOR REVISIONS before publishing. The authors have performed an interesting work but they should do some changes to improve its quality.

Author Response

Response to Reviewer 2 Comments

I like the effort of the authors to compile the available information about safety and toxicity of these phenolic compounds, which have been reported to exert many biological activities. Lately, numerous scientific works have stated that compounds extracted from macroalgae may be use in different industrial applications, thus, evaluate their safety is very necessary.

In the following paragraphs, I will provide clear information in order to improve the manuscript. In general, the English level is good and I cannot see any discrepancies. I have detected that italics is missing in many species names throughout the manuscript, so I suggest the authors to carefully check these mistakes.

Introduction:

Point 1: The first section is not fluid; the authors used too much “phlorotannins”.

Response 1: Thank you for your comment. In response to your comment, we have revised to “Phlorotannins are polyphenols found in brown seaweeds, consisting of phloroglucinol (Figure 1) (1,3,5-trihydroxybenzene) units that are bonded to each other by different pathways. They are found in the range of 126–650 kDa, and their concentration in dried brown seaweeds varies from 0.5% to 2.5% [1-2].”

Point 2: Line 47: “higher phenolic compounds”??

Response 2:  In response to your comment, we have revised to “Among seaweeds, brown seaweeds including E. cava, have been reported to produce higher concentrations of phlorotannins than other marine phenolic compounds [12].”

Point 3: Line 48: In my opinion, it would be interesting to explain a bit more the function of phlorotannins in brown seaweeds.

Response 3:

In response to your comment, we have added “In brown seaweeds, phlorotannins act as UV protectors and antioxidants, prevent stress and herbivory, and play an important role in the structure of the cell wall [14-16].” in revised manuscript.

Point 4: Lines 51-60: Is there a reason why the authors focus on the species Eisenia bicyclis, E. arborea, E. cava, E. kurome, E. stolonifera, Pelvetia siliquosa, and Ishige okamura? There are phlorotannins extracted from other brown macroalgae which have been reported to exert biological properties.

Response 4:

In response to your comment, we have revised to “Eisenia bicyclis, E. arborea, E. cava, E. kurome, Ecklonia stolonifera, Pelvetia siliquosa, and Ishige okamurae, as well as from the genera Cystophora and Fucus, contain phlorotannins that possess antidiabetic, antioxidant, antitumor, anti-inflammatory, and anticancer properties [27–28]”

Point 4: Line 51: and

In my opinion, this section is not enough focused regarding the objective presented by the authors. It is too general. Authors mentioned have mentioned food, pharmaceutical and cosmetic products in the abstract, but in the introduction they do not mention the need to evaluate the toxicity of phlorotannins for such applications. In addition, I would also suggest to remark the possibilities offered by phlorotannins in the field of food, so that the manuscript fits better with the scope of the journal.

Response 4: In response to your comment, we have revised to “Various biological activities have been reported for phlorotannins, including anticancer [23], antioxidant [32], anti-inflammatory [33], antidiabetic, and neuroprotective [18] activities. The biological activities exhibited by these compounds suggest that they are potentially useful as new ingredients in the food, feed, and pharmaceutical industries. Nonetheless, it is important to study the safety and toxicity of these compounds before the development of new products. In this regard, a review of the safety and toxicity of phlorotannins is urgently needed in view of their functionality and potential for industrial applications. Many articles reviewing the characterization and quantitative analysis of phlorotannin compounds [36–38], as well as the functionality of phlorotannins [2,27] [39–42] have been published to date. However, although studies on the safety and toxicity of phlorotannins have been conducted, there are no review papers on this topic. Therefore, this article will review studies conducted to test the safety and toxicity of phlorotannins in various organisms.“

We added “Moreover, the application of phlorotannins as food supplements [34] and functional food ingredients [35] has been reported. As a food supplement, daily intake of phlorotannins depends on the age of the consumer. For adolescents (12-14 years of age), the maximum daily intake is 163 mg/day. For those above 14 years of age and adults, the daily intake was 230 mg/day and 263 mg/day, respectively. The European Food Safety Authority (EFSA) Panel on Dietetic Products, Nutrition and Allergies (NDA), pursuant to Regulation (EC) No. 258/97, announced that novel food supplements from phlorotannins (marketed as SeaPolynolTM) are safe for human consumption [34]” in results and discussion.

Results and discussion:

I have several comments regarding this section. Since the review aims to compile the information about the safety and toxicity.

Point 5: Is there any relationship between toxicity/safety and the structure of the phlorotannins?

Response 5: Since various biological activities of phlorotannins have been reported, no research has been conducted to address the relationship between these factors. According to the results of Arnold et al., (1998) the structure of phlorotannins determines their classification and characteristics.

Arnold, T.M.; Targett, N.M. Quantifying in situ rates of phlorotannin synthesis and polymerization in marine brown algae. J. Chem. Ecol. 1998, 24, 577–595.   

Point 6: Why authors have reviewed antiviral and antimicrobial properties? Is interesting information, but I’m not sure if it fits with the objective of the manuscript.

Response 6: In response to your comment, we have deleted section on antiviral and antimicrobial activities.

Point 7: Line 206: Himanthalia elongata.

Response 7: In response to your comment, we have revised to Himanthalia elongata.

Point 8: In Paragraph 3.5. Safety and Toxicity of Phlorotannins in Microalgae, Seaweeds and Plants, no species of seaweeds (=macroalgae) is mentioned in the text or the table, so I would recommend to change the title.

Response 8: In response to your comment, we have added seaweed data in Table 3.

Point 9: I would modify Figure 2, according with previous suggestions, and introduce it at the beginning of the section.

Response 9: In response to your comment, we have revised Figure 2 and moved into beginning of results and discussion section.

Conclusions:

Point 10: From my point of view, authors should elaborate a bit more their conclusions. They have compiled interesting information, but they do not draw elaborate conclusions.

Response 10: In response to your comment, revised to “Recent studies provided evidence that phlorotannins from brown seaweeds showed low toxicity in cell lines (15 articles); microalgae; seaweed spores; plants (3 articles); invertebrates (4 articles); animals [fish, mice, rats, and dogs (8 articles)], and humans (4 articles) at a moderate dosage. Mild side effects were recorded in humans, fish, and dogs. However, in other organisms, there was no toxicity from phlorotannins, which have various biological activities. These findings can be the basis for developing these compounds as novel functional foods, feeds, and pharmaceuticals. To date, to the best of our knowledge, no studies have been performed on the safety and toxicity of phlorotannins in aquaculture fish; livestock (chicken, duck, swine, and cattle); and companion animals (birds and cats). The safety, toxicity, and availability of phlorotannins in these organisms should be verified with further studies.”

Thank you for your valuable comments on our manuscript. We have modified our manuscript according to your suggestions and we ask that the revised manuscript be re-examined and considered for publication.

Round 2

Reviewer 2 Report

Title: Effects of Phlorotannins on Organisms: Focus on the Safety, Toxicity, and Availability of Phlorotannins

The authors have considered the comments suggested and they have done numerous changes to improve the quality of the manuscript. I have few comments:

  1. Introduction

From my point of view, the changes suggested in this section have been carried out correctly and the introduction has been improved.

Line 62: Please, uniform the abbreviations of the genera.

Line 63: genera Cytospora and Fucus should be in italics.

  1. Results & discussion

The authors have also accomplished the comments suggested in this section.

In my opinion, Lines 148-156 present interesting information, but I would suggest to move it to section 3.6. Safety and toxicity of phlorotannins in humans.

Author Response

Response to Reviewer 2 Comments

The authors have considered the comments suggested and they have done numerous changes to improve the quality of the manuscript. I have few comments:

Introduction:

From my point of view, the changes suggested in this section have been carried out correctly and the introduction has been improved.

Point 1: Line 62: Please, uniform the abbreviations of the genera.

Response 1: Thank you for your comment. In response to your comment, we have revised the abbreviations of the genera.

Point 2: Line 63: genera Cytospora and Fucus should be in italics.

Response 2: In response to your comment, we have revised the genera Cytospora and Fucus in italics.

Results and discussion:

The authors have also accomplished the comments suggested in this section.

Point 3: Lines 148-156 present interesting information, but I would suggest to move it to section 3.6. Safety and toxicity of phlorotannins in humans.

Response 3: In response to your comment, we have moved into section 3.6. Safety and toxicity of phlorotannins in humans.

Thank you for your valuable comments on our manuscript. We have modified our manuscript according to your suggestions and we ask that the revised manuscript be re-examined and considered for publication.